# Pass@k Metric for RLVR: A Diagnostic Tool of Exploration, But Not an Objective

## Abstract

The ability of Large Language Models (LLMs) to perform complex, multi-step reasoning is a central focus of modern AI research. To evaluate and enhance this capability, the `pass@k` metric, which measures the probability of obtaining at least one correct solution in $k$ independent samples, has received significant attention. Its intuitive appeal has led to its adoption not only as an evaluation standard but also as a direct optimization objective in reinforcement learning. In this paper, we analyze the `pass@k` objective, derive its gradient, and demonstrate that it is fundamentally a per-example positive reweighting of the simpler `pass@1` objective. Our analysis reveals that the `pass@k` objective provides a vanishing learning signal in situations where exploration is most critical. We further analyze the dynamics of "exploration collapse", showing that as the policy concentrates probability mass, the gap between `pass@k` and `pass@1` diminishes. We conclude that while `pass@k` is a useful diagnostic tool, it may be an unsuitable direct objective for optimization. Instead, mechanisms explicitly encouraging efficient exploration could offer a more effective path forward for reinforcement learning in reasoning tasks.

## 1 Introduction

Large Language Models (LLMs) have demonstrated remarkable capabilities across a wide range of natural language tasks (2). A frontier of current research is pushing these models beyond simple pattern matching toward complex, multi-step reasoning. Techniques such as Chain-of-Thought (CoT) prompting (12) have shown that eliciting intermediate reasoning steps significantly improves performance on arithmetic, commonsense, and symbolic reasoning tasks.

A natural extension of this paradigm is to generate multiple reasoning trajectories and select the best one. Methods like Self-Consistency (11) leverage this idea by sampling multiple diverse reasoning paths and aggregating the results. This underscores a crucial insight: a model's capacity is defined not merely by its ability to produce a single correct answer, but by its ability to explore a distribution of potential solutions that contains correct answers.

To quantify this exploratory capacity, the `pass@k` metric (4) has been widely adopted. `pass@k` measures the probability that at least one of $k$ i.i.d. samples from a model yields a correct solution. Given its utility as an evaluation metric, it is tempting to use `pass@k` directly as a reward signal in a reinforcement learning (RL) loop. The intuition is that by rewarding the model for succeeding in at least one of $k$ attempts, the RL process will be incentivized to broaden its solution search space.

In this paper, we challenge this intuition. We show that optimizing for `pass@k` reduces to reweighting per-example `pass@1` gradients by a positive, success-dependent scalar. This coupling reveals two potentially critical issues:

1. When the model consistently fails to find a correct solution (`pass@1` $\approx 0$), the empirical gradient of the `pass@k` objective effectively vanishes. Thus, `pass@k` fails to provide a learning signal exactly when it is most needed.

2. As the policy improves and concentrates mass on a solution, `pass@k` converges to `pass@1`, rendering the computational cost of generating $k$ samples redundant during training.

We conclude that `pass@k` should be regarded as a *diagnostic* of a model's reasoning diversity rather than a *prescription* for optimization.

## 2 Background and Related Work

### 2.1 Reinforcement Learning Formulation for LLMs

We consider the standard autoregressive language modeling setting, where a Large Language Model (LLM) functions as a stochastic policy $\pi_\theta$, parameterized by weights $\theta$. Given a context or prompt $x$ drawn from a distribution $\mathcal{D}$, the model generates a sequence of tokens $y = (a_1, a_2, \ldots, a_T)$, where each token $a_t$ is selected from a vocabulary $\mathcal{V}$.

The generation process can be formulated as a Markov Decision Process (MDP) where:

- The state $s_t$ at time step $t$ consists of the prompt and the history of generated tokens: $s_t = (x, a_1, \ldots, a_{t-1})$.

- The action $a_t$ is the next token generated by the model.

- The policy $\pi_\theta(a_t|s_t)$ defines the probability distribution over the vocabulary given the current state.

- The transition is deterministic; the next state is simply the concatenation of the current state and the chosen action: $s_{t+1} = (s_t, a_t)$.

The joint probability of generating a full response $y$ given prompt $x$ is the product of the conditional probabilities:

$$\pi_\theta(y|x) = \prod_{t=1}^{T} \pi_\theta(a_t|x, a_{<t}). \tag{1}$$

In the reinforcement learning framework, we assume the existence of a reward function $r(x, y) \in \mathbb{R}$, which evaluates the quality of the completed sequence $y$ given the prompt $x$. In Reinforcement Learning from Human Feedback (RLHF) (15; 7), this reward model is learned from human preferences. The reward is typically sparse, provided only at the end of the generation (terminal reward). The objective of the RL process is to learn policy parameters $\theta$ that maximize the expected reward over the data distribution:

$$J(\theta) = \mathbb{E}_{x \sim \mathcal{D}} \mathbb{E}_{y \sim \pi_\theta(\cdot|x)}[r(x, y)]. \tag{2}$$

Standard RL algorithms used for LLMs estimate the gradient of this objective using samples generated by the current policy.

### 2.2 RL with Verifiable Rewards

To enhance the reasoning ability of LLMs, Reinforcement Learning with Verifiable Rewards (RLVR) replaces the learned reward model with a deterministic, programmatic verifier $V(x, y) \in \{0, 1\}$ that checks the correctness of final answers (e.g., unit tests for code or exact matches for math).

The standard single-try objective ($k = 1$) is:

$$J_1(\theta) = \mathbb{E}_{x \sim \mathcal{D}} \left[ \mathbb{E}_{y \sim \pi_\theta(\cdot|x)} \left[ V(x, y) \right] \right]. \tag{3}$$

When allowing for $k$ attempts per prompt, the objective becomes maximizing the probability that at least one sample is correct:

$$J_k(\theta) = \mathbb{E}_{x \sim \mathcal{D}} \left[ \mathbb{E}_{y_{1:k} \sim \pi_\theta(\cdot|x)} \left[ 1 - \prod_{i=1}^{k} \big( 1 - V(x, y_i) \big) \right] \right]. \tag{4}$$

### 2.3 Pass@k in RLVR

The `pass@k` metric originated in program synthesis evaluation for unit-tested code (4) and has become standard in verifiable domains. Most RLVR literature reports both `pass@1` and `pass@k` on benchmarks such as HumanEval/MBPP for code (4; 1) and GSM8K/MATH for mathematics (5; 6).

While direct gradient optimization of Eq. 4 is an attractive theoretical target, a parallel line of research optimizes `pass@k` implicitly via iterative data generation. Methods such as STaR (14) and Rejection Sampling Fine-Tuning (RFT) (9) generate $k$ samples, filter for correctness, and fine-tune on the correct trajectories. However, recent analysis suggests that standard RLVR often narrows the exploration space rather than expanding it (13).

Emerging research has begun to explicitly address the limitations of standard RLVR in optimizing `pass@k`, confirming the theoretical concerns we raise regarding mode collapse and exploration. Yue et al. (13) conduct a critical examination of whether RLVR incentivizes reasoning capacity. They find that under the standard implementation (that restricts the exploration space of RL), while RLVR improves sampling efficiency (boosting `pass@1`), it often reduces the model's reasoning boundary compared to the base model. Specifically, they observe that the base model often outperforms the RL-trained model at large $k$, suggesting that the implementation of RLVR induces a collapse in diversity.

To counter these limitations, Walder and Karkhanis (10) propose Pass@K Policy Optimization (PKPO). They argue that standard RL rewards samples independently, optimizing `pass@1` at the expense of collective utility, and derive unbiased estimators for the `pass@k` gradient that consider the set of $k$ samples jointly. Similarly, Chen et al. (3) introduce `pass@k` training, deriving an analytical advantage function that balances exploration and exploitation. Addressing the mechanism of collapse, Peng et al. (8) introduce SimKO. They identify "probability concentration" on top-1 tokens as the primary cause of reduced `pass@k` performance. To mitigate this, SimKO employs asymmetric updates: boosting the probability of the top-$K$ candidates for correct responses while applying stronger penalties to the top-1 candidate for incorrect responses.

Despite these advances, we argue that making `pass@k` the direct optimization objective remains problematic. We posit that high `pass@k` is a downstream consequence of effective exploration, rather than a causal driver of learning performance that should be directly optimized.

## 3 Analysis of the Pass@k Objective

In this section, we formalize the `pass@k` objective and derive its gradient. This allows us to analyze the learning signal it provides to the model during optimization.

### 3.1 The Gradient of the Pass@k Objective

Let $J_1(x; \theta)$ be the probability that a single trajectory is correct. The `pass@k` objective is defined functionally as:

$$J_k(x; \theta) = 1 - (1 - J_1(x; \theta))^k. \tag{5}$$

Applying the chain rule to Equation equation 5 for a fixed input $x$:

$$\nabla_\theta J_k(x; \theta) = \nabla_\theta \left[ 1 - (1 - J_1(x; \theta))^k \right]$$
$$= k (1 - J_1(x; \theta))^{k-1} \cdot \nabla_\theta J_1(x; \theta).$$

**Theorem 3.1** (Per-example gradient relation)*. For any fixed input $x$, the gradient of the `pass@k` objective is a scalar multiple of the gradient of the `pass@1` objective:*

$$\nabla_\theta J_k(x; \theta) = \alpha_k(x, \theta) \cdot \nabla_\theta J_1(x; \theta), \tag{6}$$

*where the scaling factor is*

$$\alpha_k(x, \theta) = k (1 - J_1(x; \theta))^{k-1} \in [0, k]. \tag{7}$$

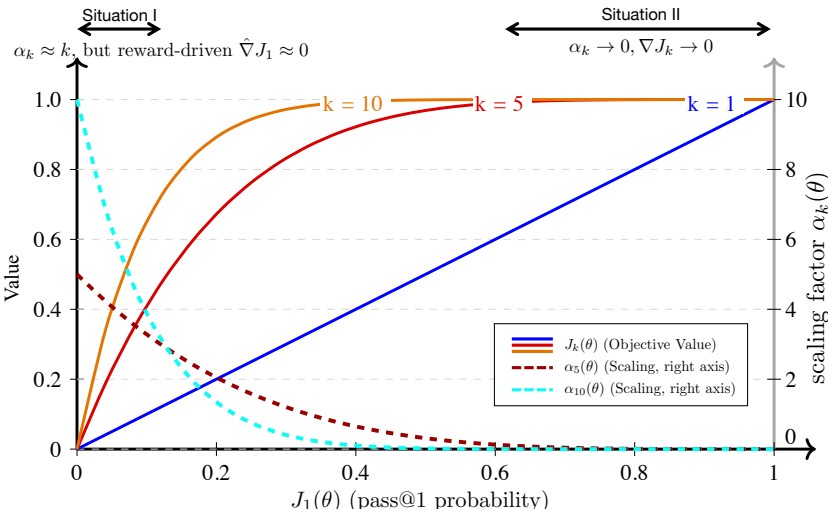

Figure 1: Visualization of $J_k(\theta)$ vs. $J_1(\theta)$ (solid) and the scaling factor $\alpha_k(\theta)$ (dashed, scaled by 0.5 for visualization).

This result highlights a fundamental property: optimizing `pass@k` does not introduce a new search direction in the parameter space. Instead, it dynamically reweights the standard `pass@1` gradient based on the model's current performance.

The behavior of `pass@k` as an optimization objective is entirely dictated by the scaling factor $\alpha_k(x, \theta)$. We discuss two distinct situations below, which are depicted in Figure 1.

**Situation I, low $J_1$:** Consider the case where the policy assigns negligible probability to correct trajectories, i.e., $J_1(x; \theta) \approx 0$. As $J_1 \to 0$, the scaling factor approaches its maximum: $\lim_{J_1 \to 0} \alpha_k = k$. Theoretically, the gradient signal should be amplified. Despite the high theoretical scaling factor, the gradient $\nabla_\theta J_1(x; \theta)$ must be estimated via sampling (e.g., REINFORCE). If the model fails to sample any correct solutions (which is probable when $J_1 \approx 0$), the empirical gradient estimate is zero. Consequently, the `pass@k` objective fails to provide a learning signal exactly when the model needs it most, regardless of the multiplier $k$.

**Situation II, high $J_1$:** Consider the case where the model becomes reliable, i.e., $J_1(x; \theta) \to 1$. For any $k > 1$:
$$\lim_{J_1 \to 1} \alpha_k(x, \theta) = 0.$$

Consequently, $\nabla_\theta J_k(x; \theta) \to \mathbf{0}$. Intuitively, if success on the first attempt is nearly certain, the condition of "succeeding in at least one of $k$ attempts" is already satisfied. The objective saturates, and the learning signal vanishes. This creates a scenario where the model has no incentive to further improve or explore, as the marginal utility of additional correct samples under `pass@k` is zero.

### 3.2 Convergence of Pass@k to Pass@1 Under Exploration Collapse

We further analyze the dynamics of iterative reinforcement learning when applied to reasoning tasks. We demonstrate that as the policy improves, the diversity of generated samples decreases. We show that as the policy concentrates probability mass on the discovered mode, the *marginal utility of taking k samples diminishes*.

### 3.3 Exploration Vanishing and Diminishing Returns of $k$

Iterative Reinforcement Learning relies on estimating gradients using samples generated by the current policy $\pi_t$. We demonstrate that if a mode of the optimal distribution falls below a probability threshold $\epsilon$, the

sampling process is statistically likely to miss it entirely. Consequently, the policy concentrates exclusively on the already discovered mode. This concentration causes the performance of multi-sample generation (`pass@k`) to collapse toward the performance of a single sample (`pass@1`).

**Theorem 3.2** (Exploration Vanishing). *Let $Y^*$ be the set of optimal trajectories containing two disjoint modes, $M_1$ (discovered) and $M_2$ (undiscovered). Let $k$ be the number of samples drawn at step $t$. If the current policy satisfies $\pi_t(M_2) < \epsilon$, then:*

1. *The probability of discovering $M_2$ is bounded by approximately $k\epsilon$.*

2. *As $\epsilon \to 0$, the policy updates will monotonically increase the probability mass of $M_1$, i.e., $\pi_{\theta_{t+1}}(M_1) \geq \pi_{\theta_t}(M_1)$.*

3. *As $\pi_t(M_1) \to 1$, the performance gap between `pass@k` and `pass@1` vanishes.*

*Proof.* The proof proceeds in three steps: establishing the probability of missing the second mode, deriving the concentration of mass on the first mode, and calculating the resulting collapse of the `pass@k` metric.

Step 1: Probability of missing $M_2$. Let $S_t = \{y_1, \ldots, y_k\}$ be $k$ i.i.d. samples drawn from $\pi_t$. The probability that a single sample misses $M_2$ is $1 - \pi_t(M_2) > 1 - \epsilon$. The probability that *all* $k$ samples miss $M_2$ is:

$$P(S_t \cap M_2 = \emptyset) > (1 - \epsilon)^k.$$

Conversely, the probability of discovering the mode is upper bounded by the first-order Taylor expansion:

$$P(\text{discovery}) = 1 - (1 - \pi_t(M_2))^k < 1 - (1 - \epsilon)^k \approx k\epsilon. \tag{8}$$

Thus, when $\epsilon \to 0$, the gradient estimator will almost surely contain no signal from $M_2$.

Step 2: Concentration on $M_1$. We analyze the change in probability mass $p_t = \pi_{\theta_t}(M_1)$ . The optimizer updates parameters $\theta$ to maximize the log-likelihood of the observed batch $S_t$. The general gradient ascent update rule is:

$$\theta_{t+1} = \theta_t + \eta \sum_{x \in S_t} \nabla_\theta \log \pi_{\theta_t}(x) \tag{9}$$

where $\eta$ is the learning rate.

When $\epsilon \to 0$, the likely event occurs where $S_t \cap M_2 = \emptyset$. This implies that all observed samples $x \in S_t$ belong to the mode $M_1$. Therefore, maximizing the likelihood of these individual samples is equivalent to maximizing the aggregate probability mass of $M_1$. We approximate the sum of sample gradients as the gradient of the log-probability of the set $M_1$:

$$\sum_{x \in S_t} \nabla_\theta \log \pi_{\theta_t}(x) \approx \nabla_\theta \log p_t = \frac{\nabla_\theta p_t}{p_t} \tag{10}$$

Substituting 10 into 9, the specific parameter update becomes:

$$\theta_{t+1} = \theta_t + \eta \frac{\nabla_\theta p_t}{p_t} \tag{11}$$

To find the new probability mass $p_{t+1} = \pi_{\theta_{t+1}}(M_1)$, we apply a first-order Taylor expansion around $\theta_t$:

$$p_{t+1} \approx p_t + (\nabla_\theta p_t)^\top (\theta_{t+1} - \theta_t) \tag{12}$$

$$= p_t + (\nabla_\theta p_t)^\top \left( \eta \frac{\nabla_\theta p_t}{p_t} \right) \tag{13}$$

$$= p_t + \frac{\eta}{p_t} (\nabla_\theta p_t)^\top (\nabla_\theta p_t) \tag{14}$$

$$= p_t + \frac{\eta}{p_t} \|\nabla_\theta p_t\|^2 \tag{15}$$

Since the learning rate $\eta > 0$, the probability $p_t > 0$, and the squared norm $\|\nabla_\theta p_t\|^2 \geq 0$, it follows that:

$$p_{t+1} \geq p_t, \quad \text{i.e.,} \quad \pi_{\theta_{t+1}}(M_1) \geq \pi_{\theta_t}(M_1) \tag{16}$$

This confirms that when the correct answer is not observed, the model reinforces the probability of the mode $M_1$.

Step 3: Vanishing `pass@k` gap. We define the gap as the marginal benefit of drawing $k$ samples versus 1 sample. Assuming correctness is defined by falling into the dominant mode $M_1$:

$$\text{pass@1} = p_t, \tag{17}$$
$$\text{pass@k} = 1 - (1 - p_t)^k. \tag{18}$$

The gap is given by $\Delta_t(k) = \text{pass@k} - \text{pass@1} = 1 - (1 - p_t)^k - p_t$. As the policy concentrates (from Step 2), let $p_t = 1 - \delta$ for some small residual exploration mass $\delta > 0$. Substituting this into the gap equation:

$$\Delta_t(k) = 1 - \delta^k - (1 - \delta) = \delta - \delta^k. \tag{19}$$

Since $\delta < 1$ and $k \geq 1$, as the policy updates and $\delta \to 0$, the term $\delta - \delta^k$ approaches 0. Therefore, as the RL algorithm iterates and concentrates probability mass on $M_1$, `pass@k` provides diminishing returns and converges to `pass@1`. $\qquad\square$

This theoretical result aligns with the empirical findings of Peng et al. (8), who observed that stronger over-concentration correlates with worse `pass@k` performance. This confirms that exploration collapse directly undermines the utility of the $k$ parameter.

## 4 Discussion and Conclusion

In this work, we conducted a formal mathematical analysis of the `pass@k` metric to evaluate its suitability as an optimization objective. Our findings expose a fundamental dichotomy between the metric's utility as a diagnostic tool and its efficacy as a training target.

On one hand, the discrepancy between `pass@1` and `pass@k` serves as a useful indicator of model calibration. A scenario where `pass@1` is low while `pass@k` remains high indicates that the model's latent distribution covers the correct solution space, yet the policy lacks the confidence to assign high probability to these modes. In this context, `pass@k` effectively measures the potential of the model under repeated sampling.

On the other hand, our analysis shows that using `pass@k` as a surrogate objective for standard policy gradient methods is theoretically unsound. We demonstrated that the gradient of the expected `pass@k` is strictly collinear with that of `pass@1`, offering no distinct optimization signal. Furthermore, we showed that as a policy concentrates mass—whether through training convergence or temperature reduction—the `pass@k` metric mathematically converges to `pass@1`. This renders the computational expense of generating $k$ samples redundant during training in the situation where performance gains are sought. Moreover, as corroborated by recent empirical studies (13; 8), standard RLVR tends to induce entropy collapse, actively eroding the sample diversity required for `pass@k` to be meaningful.

To effectively optimize for `pass@k`, algorithms must transcend the standard assumption of independent sample evaluation. But ultimately, we advocate for retaining `pass@k` as an inference-time diagnostic, while adopting advanced exploration mechanisms that explicitly target diversity and collective utility for training.

## Broader Impact Statement

This work is primarily a theoretical analysis of optimization objectives in reinforcement learning for LLMs. As such, its immediate impact is methodological.

## Acknowledgments

Gemini 2.5 Pro was employed to improve the language of this paper.

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
