# OpenReview forum: "Pass@k Metric for RLVR: A Diagnostic Tool of Exploration, But Not an Objective"
_TMLR — Rejected by TMLR_

### Review · Reviewer_pitM · 2025-12-30

**Summary Of Contributions:**

**Summary.**
This paper analyzes the pass@k metric in RLVR and argues that pass@k should be viewed primarily as a diagnostic of exploration rather than a training objective. It shows that, for the idealized objectives, the gradient of pass@k is collinear with (and merely a scalar reweighting of) the gradient of pass@1. It then discusses two regimes (low pass@1 and high pass@1) and presents an “exploration collapse” analysis (Theorem 3.2) intended to explain why pass@k tends to converge toward pass@1 as policies concentrate probability mass.

**Key strengths.**

- The gradient relationship between ideal pass@k and pass@1 is correct and clearly presented
- The topic is timely and relevant
- The paper is concise and easy to read

**Key weaknesses.**

- The theoretical analysis contains several key issues (see my detailed comments below)
- Missing relevant related work (e.g., [1,2])
- No empirical study; the paper reads more like a short theory note than a full research contribution

References

[1] Chow, Yinlam, et al. "Inference-aware fine-tuning for best-of-n sampling in large language models." ICLR 2025

[2] Tang, Yunhao, et al. "Optimizing language models for inference time objectives using reinforcement learning." ICML 2025

**Audience:**

Yes

**Audience Explanation:**

The question of whether pass@k should be optimized (and how) is relevant to research in RL, LLMs, and beyond. Even a limited theoretical clarification can be valuable as a conceptual check on common intuitions. The gradient relationship and the broader discussion of exploration collapse are likely to interest researchers thinking about objective design and evaluation in this space.

**Claims And Evidence:**

No

**Claims Explanation:**

The relationship between $J_1$ and $J_k$ is correct. However, the evidence does not convincingly support the stronger conclusion that pass@k is unsuitable as an optimization objective, due to three issues in the theoretical analysis.

- **“Situation I (low $J_1$)”: mixed level of abstraction**.
$J_1$ and $J_k$ are ideal population-level objectives. The paper first derives a relationship between their gradients, and then argues that when pass@1 is small, *empirically* the gradient estimate is zero because one is likely to sample only incorrect trajectories, so the pass@k gradient is also zero despite the large scaling factor. This mixes an ideal gradient identity with a specific finite-sample, estimator-dependent behavior (e.g., REINFORCE or PPO). It is incorrect to multiply a population-level scalar by a finite-sample zero and draw conclusions about the population gradient. Moreover, the claim that the "empirical gradient is zero'' is algorithm-specific and depends on baseline choice and in-group normalization.

- **“Situation II (high $J_1$)” and the first part of Theorem 3.2: correct but not informative limit arguments.** The paper argues that as pass@1 approaches 1, the scaling factor goes to 0 and hence the pass@k gradient vanishes; similarly, the first part of Theorem 3.2 uses $\varepsilon \to 0$ to argue a "vanishing signal''. These statements are mathematically correct but not particularly diagnostic of pass@k, because vanishing gradients near saturation are generic for bounded success-probability objectives. Moreover, taking $\varepsilon \to 0$ collapses exactly the finite regime where pass@k can differ meaningfully from pass@1 and where its practical relevance would need to be assessed.

- **The final step of Theorem 3.2: a trivializing assumption.** In the last part of the proof, the paper assumes that correctness is defined by membership in the dominant mode. Under this assumption, the statement that pass@k converges to pass@1 as the policy mass concentrates on that mode becomes essentially trivial. This substantially weakens the theorem’s role as evidence for the paper’s broader narrative about exploration collapse.

**Requested Changes:**

- Resolve the mixed level of abstraction in the theoretical analysis, especially in “Situation I”. The paper should either stay at the level of ideal objectives or explicitly analyze the practical surrogate objectives and estimators (e.g., PPO/GRPO) used in RLVR, rather than combining the two without justification

- Revise and improve the arguments in “Situation II” and first part of Theorem 3.2 to avoid relying on uninformative limit arguments

- Revise and improve the final part of Theorem 3.2, as the assumption that correctness is defined by the dominant mode trivializes the conclusion

- Add missing related work

- Add some empirical results to support the theoretical claims

---

### Review · Reviewer_CTnQ · 2026-01-07

**Summary Of Contributions:**

Authors argue that pass@k metric is no good metric where exploration is most critical in terms of reinforcement learning refinement for LLMs. With theoretical analysis, it is shown that pass@k cannot resolve the case where pass@1 is ineffective either, proving that it cannot be a good substitution.

**Audience:**

Yes

**Audience Explanation:**

Overall the paper is somewhat like a preliminary work rather than a full paper for journal. Preliminary analysis and observation are presented but deeper empirical experiments and broader analysis are missing.

**Broader Impact Concerns:**

No further ethical considerations.

**Claims And Evidence:**

Yes

**Claims Explanation:**

The paper has clear theoretical evidence to support the argument raised.

As authors discuss in the discussion and conclusion section: "A scenario where pass@1 is low while pass@k remains high indicates that the model’s latent distribution covers the correct solution space", pass@k can sometimes work as a more efficient metric. If we look at figure 1 and equation5, pass@k led reward gradient has higher order which means that pass@k can potentially reduce the number of expensive full-length evaluations. Can authors have more discussion on this?

 Although the theoretical analysis is sufficient for the argument, to make the paper more self-contained and be more suitable for wider audience, I think empirical experiments (e.g. math500, aime24 etc.) will make the paper stronger.

I also have a questions
There are a lot of scenarios of RL for LLM. Does this manuscript focus on RLHF? (though RLHF is still very broad).

**Requested Changes:**

Apart from my questions and suggestions previously. I would recommend authors to make the paper more self-contained including proper experiment evaluation supporting the arguments and formal proof for the mathematical analysis.

---

### Review · Reviewer_TQGZ · 2026-01-20

**Summary Of Contributions:**

This paper investigates the pass@k metric and specifically its use as an optimization objective, which is a recent trend in Reinforcement Learning with Verifiable Reward (RLVR). The authors prove that the gradient of the pass@k objective a scalar multiple of the pass@1 gradient, implying that it fails to introduce new search directions during optimization beyond those already induced by pass@1. The analysis identifies two specific failure modes: (1) that the learning signal vanishes when exploration is most needed (i.e. when pass@1 probability is low and thus, despite amplifying the pass@k gradient the model is unlikely to sample trajectories yielding non-negative signal); and (2) when the model has already reached high reliability and thus fails to explore since it already succeeds with high likelihood on a first attempt (Figure 1). The study finally characterizes a phenomenon the authors call "exploration collapse" in which a policy concentrates on discovered, causing pass@k to converge toward pass@1 and thus rendering the computational cost of multiple samples redundant. The final conclusion of the authors is that tasks tends to erode sample diversity, the authors argue that *pass@k is better suited as a diagnostic tool than a training target*.

The paper has the following strong points:
+ **Timeliness**: RLVF is currently a hot topic, with very recent works specifically attempting to directly optimize pass@k and mitigate its empirically demonstrated pitfalls. This paper provides a convincing argument as to why these pitfalls occur.
+ **Clarity**: The paper is tight, concise, and correct. The theoretical arguments and proofs, as well as the motivational analysis leading the reader through the line of argumentation, is extremely clear and intuitive. The main results (Theorems 3.1 and 3.2) are convincing.
+ **Scope**: The paper does one thing and does it well. It doesn't try to do more than is necessary, and the purely theoretical results are left to stand on their own (which I think they do).

The only weakness I can see with the work is the simplified version of RLVR upon which Theorem 3.2 is based. In particular, it is unclear whether more complex RLVR approaches, ones not based on vanilla policy gradient, might violate the assumptions of the proof.

**Audience:**

Yes

**Audience Explanation:**

RLVR is currently of significant interest to the broader ML community, with multiple recent papers at top conferences (ICML and NeurIPS) and several currently under review at ICLR which specifically attempt to optimize the pass@k metric. I expect the paper to be of interest to anyone currently working on RLVR.

**Broader Impact Concerns:**

The paper is purely theoretical. If I had to invent something, I might suggest investigating whether expanding reasoning boundaries in RLVR leads to increased risk of reward hacking, which then would likely break alignment with human expectations.

**Claims And Evidence:**

Yes

**Claims Explanation:**

The theoretical treatment is top notch and reads extremely well. I do not detect any errors of bugs in clarity in the paper.

**Requested Changes:**

Some of the papers cited as arXiv have been published (e.g. [10] at NeurIPS 2025 and [13] at ICML 2025). All bibliographic entries should be carefully checked for published versions.

Subsection 3.2 seems a bit out of place. Perhaps the authors intended to start a new section with this paragraph, which makes more sense because it is more of an introduction to the content of subsections 3.3.

---

### Decision · Action_Editor_WXKs · 2026-03-02

**Recommendation:** Reject

**Additional Comments:**

This paper provides a theoretical analysis of the pass@k objective in RLVR. The idea is to show that this objective is just a reweighting that does not really help in the cases where exploration is necessary.

The basic idea is interesting—and timely given how popular RLVR methods for reasoning are. However, reviewers also brought up a bunch of relevant questions that have not yet been answered. First, there are a number of issues: for example, mixing finite-sample (per-prompt) versus average dataset or population-level results. Similarly, it’s not clear how Step 2 in Thm 3.2 works (or the exact setup for it). Second, the assumptions made are pretty strong, which might limit the results. Third, it would be nice to support the theoretical results with empirical validation. This is not strictly required but would be extremely useful to see.

There’s clearly some interesting aspects for this paper, but it is not quite ready yet.

**Audience:**

Yes

**Audience Explanation:**

This paper focuses on RLVR and has a wide potential audience.

**Claims And Evidence:**

No

**Claims Explanation:**

There are claims that may have some flaws (or at least require some clarification).

**Resubmission Of Major Revision:**

The authors may consider submitting a major revision at a later time.